# Presepsin as a Potential Prognostic Marker for Sepsis According to Actual Practice Guidelines

**DOI:** 10.3390/jpm11010002

**Published:** 2020-12-22

**Authors:** Alice Nicoleta Drăgoescu, Vlad Pădureanu, Andreea Doriana Stănculescu, Luminița Cristina Chiuțu, Dan Nicolae Florescu, Ioana Andreea Gheonea, Rodica Pădureanu, Alex Stepan, Costin Teodor Streba, Andrei Ioan Drocaș, Adriana Mihaela Ciocâlteu-Ionescu, Valeriu Marin Șurlin, Octavian Petru Drăgoescu

**Affiliations:** 1Department of Anesthesiology and Intensive Care, University of Medicine and Pharmacy of Craiova, 200349 Craiova, Romania; alice.dragoescu@yahoo.com (A.N.D.); luminita.chiutu@gmail.com (L.C.C.); 2Department of Internal Medicine, University of Medicine and Pharmacy of Craiova, 200349 Craiova, Romania; 3Department of Gastroenterology, University of Medicine and Pharmacy of Craiova, 200349 Craiova, Romania; nicku_dan@yahoo.co.uk (D.N.F.); adriana_ciocalteu@yahoo.com (A.M.C.-I.); 4Department of Radiology and Medical Imaging, University of Medicine and Pharmacy of Craiova, 200349 Craiova, Romania; iagheonea@gmail.com; 5Department of Biochemistry, University of Medicine and Pharmacy of Craiova, 200349 Craiova, Romania; zegheanurodica@yahoo.com; 6Department of Pathology, University of Medicine and Pharmacy of Craiova, 200349 Craiova, Romania; astepan76@yahoo.com; 7Department of Research Methodology, University of Medicine and Pharmacy of Craiova, 200349 Craiova, Romania; costinstreba@gmail.com; 8Department of Urology, University of Medicine and Pharmacy Craiova, 200349 Craiova, Romania; andrei_drocas@yahoo.com (A.I.D.); pdragoescu@yahoo.com (O.P.D.); 9Department of Surgery, University of Medicine and Pharmacy of Craiova, 200349 Craiova, Romania; vsurlin@gmail.com

**Keywords:** sepsis, septic shock, prognosis, presepsin

## Abstract

The 2016 Surviving Sepsis Campaign guidelines define sepsis as life-threatening organ dysfunction caused by a dysregulated host response to infection. This study had the objective of assessing the efficacy of presepsin in the prognosis of sepsis. This was a single-center prospective study, performed in Craiova Emergency Hospital, that included 114 patients admitted in the Intensive Care Unit (ICU) department between 2018 and 2019 fulfilling the sepsis criteria. Including criteria were: age ≥ 18, sepsis diagnosed by the Sequential Organ Failure Assessment (SOFA) score of pulmonary, abdominal, urinary, surgical or unknown origin, as well as lactate levels ≥ 2 mmol/l and need of vasopressors for mean arterial pressure (MAP) ≥ 65 mmHg, despite adequate volume resuscitations for patients with septic shock. Patients younger than 18, pregnant, immunocompromised, or with terminal illnesses were excluded. Based on disease severity, patients were distributed into two study groups: sepsis—76 patients and septic shock—38 patients. As expected, SOFA score and most of its components (PaO_2_/FiO_2_, platelets, and Glasgow Coma Score (GCS)) were significantly modified for patients with septic shock compared to those in the sepsis group and for survivors versus non-survivors. Overall death rate was 34.2%, with a significantly higher value for patients with septic shock (55.3% vs. 23.7%, *p* = 0.035). Sepsis marker presepsin was significantly elevated in all patients (2047 ng/mL) and significantly increased for the septic shock patients (2538 ng/mL, *p* < 0.001) and non-survivors (3138 ng/mL, *p* < 0.001). A significant correlation was identified between the SOFA score and presepsin (*r* = 0.883, *p* < 0.001). The receiver operating characteristics (ROC)-Area Under Curve (AUC) analysis showed significant prognostic values for presepsin regarding both sepsis severity (AUC = 0.726, 95% confidence interval CI = 0.635–0.806) and mortality risk (AUC = 0.861, 95%CI = 0.784–0.919). In conclusion, under the revised definition of sepsis, presepsin could be a useful marker for prognosis of sepsis severity and mortality risk. Additional data are required to confirm the value of presepsin in sepsis prognosis.

## 1. Introduction

According to the 2016 Surviving Sepsis Campaign guidelines, sepsis is defined as severe organ dysfunction caused by a dysregulated host response to infection [1]. The severe organ dysfunctions are evaluated by the Sequential Organ Failure Assessment (SOFA) score with 2 points or more increase for critical patients from the Intensive Care Unit (ICU) [2]. Septic shock is a subgroup of sepsis with significant hemodynamic and metabolic malfunctions that lead to considerably increased mortality. These patients require vasopressor treatment to maintain their mean blood pressure above 65 mm Hg and have increased serum lactate (than 2 mmoL/L) after adequate fluid replacement [2]. In the ICU, sepsis is a major concern for critical patients so that its early diagnosis is crucial. Valuable biomarkers for early diagnostic of sepsis should have the subsequent features: significant and rapid rise in sepsis, swift decline after efficient treatment, and short half-life, as well as easy and straightforward determination procedure [3,4]. However, none of actual markers have the above ideal qualities. The usefulness of many different markers for early detection of sepsis and septic shock in ICU patients has been tested by numerous studies with unreliable results.

The goal of this study was to assess the efficacy of presepsin as a severity and mortality prognosis marker for sepsis patients compared to other inflammation or infection indicators, such as leucocytes, erythrocyte sedimentation rate (ESR), or C reactive protein (CRP). Presepsin has been used before in various clinical trials for sepsis evaluation due to its early diagnosis potential and suitability for antibiotic treatment efficiency evaluation, as well as assessing the prognosis of septic patients [3].

Presepsin is a 13 KDa polypeptide made by proteolytic cleavage of soluble forms of cluster of differentiation CD14 (sCD14). CD14 is present predominantly on most immune system cells, as well as cartilage, brain, liver, and intestinal cells [5].

Various studies investigating the significance of presepsin as a sepsis diagnosis marker had contradictory results. Zhang et al. showed, in a 2015 meta-analysis and systematic review, that presepsin is a useful diagnostic tool with high sensitivity and specificity [6]. Endo et al., in 2012, had different results with no significantly elevated presepsin levels found for patients with sepsis compared to those with localized infections [7]. Another randomized multicentric clinical study identified significantly higher levels of presepsin in patients with sepsis and septic shock that developed organ dysfunctions, as well as a correlation with the SOFA score [8]. Many other clinical trials investigated early diagnosis of sepsis and septic shock using other markers, like: CRP, ESR, procalcitonin, and various interleukins [9,10,11], but none of them positively identified a certain marker for accurate early diagnosis.

## 2. Materials and Methods

This was a single-center study performed in our teaching hospital, Craiova Emergency Hospital. It was an observational, prospective study that included 114 patients admitted in the ICU department between 2018 and 2019 for sepsis or septic shock. Including criteria for the study were: age ≥ 18, sepsis diagnosed by the SOFA score of pulmonary, abdominal, urinary, and surgical or unknown origin, as well as lactate levels ≥ 2 mmoL/L and need of vasopressors for mean arterial pressure (MAP) ≥65 mmHg, despite adequate volume resuscitations for patients with septic shock, while patients younger than 18, pregnant, immunocompromised, or with terminal illnesses were excluded. Based on disease severity, patients were divided into two study groups: sepsis—76 patients and septic shock—38 patients.

After ICU admission, signed informed consent (by patient or close relatives if patient was unconscious or not able) was obtained. Medical history, signs, and symptoms, as well as vital signs, were collected, and complete clinical examination performed, along with Glasgow Coma Score (GCS) evaluation. Routine venous blood sampling was drawn by venipuncture at the time of ICU admission for hematology (including white blood cells—WBC), biochemistry, coagulation, and microbiology, as well as erythrocyte sedimentation rate (ESR), C-reactive protein (CRP), serum lactate, arterial blood gases (ABG), and presepsin. Microbiological examination included two pairs of blood cultures (aerobic and anaerobic) and cultures from broncho-alveolar lavage fluid, sputum, urine, cerebrospinal fluid, and surgical wounds were sampled before antibiotic treatment. Radiological imaging included X-ray, ultrasound, and computed tomography [2].

Among enrolled patients, data on SOFA score components were collected and graded: PaO_2_/FiO_2_, blood platelets, total bilirubin, Glasgow Coma Score, serum creatinine, and mean arterial pressure.

In our hospital, presepsin was measured using the ICU bedside method by PATHFAST^TM^ Immunoassay Analytical System, Mitsubishi Chemical Corporation, Tokyo, Japan (chemiluminescent enzyme immunoassay—CLEIA, for the quantitative measurement of the presepsin levels in whole blood or plasma), using ethylenediaminetetraacetic acid (EDTA) whole blood and PATHFAST Presepsin test kits PF1201-K (LSI Medience Corporation, provided by Mitsubishi Chemical Europe GmBH, Düsseldorf, Germany). Presepsin values are usually interpreted as follows [12]:-Presepsin < 200 pg/mL—sepsis excluded;-Presepsin < 300 pg/mL—systemic infection improbable;-Presepsin < 500 pg/mL—sepsis probable;-Presepsin < 1000 pg/mL—significant risk of severe sepsis;-Presepsin ≥ 1000 pg/mL—high risk of severe sepsis/septic shock equivalent to SOFA ≥ 8.

All statistical tests and analysis were performed using the MedCalc^®^ statistical software version 18.11.6 and SciStat^®^ online platform. Data distribution was evaluated by the Kolomogorov–Smirnov test. Parameters with normal distribution were compared by the Student *t*-test, while Mann–Whitney *U*-test was used as a non-parametric test. Quantitative parameters were presented as mean ± SD (standard deviation) or median and 25%/75% quartile range, while other qualitative parameters were evaluated by Fisher’s exact test or Chi^2^ test. Correlations between study parameters were evaluated by Pearson *r* correlation. Receiver operating characteristics (ROC) curve analysis was performed with Area Under Curve (AUC) calculation and parameter cut-off values, as well as AUC comparison. *p*-value of 0.05 was used for statistical significance.

## 3. Results

Our patient sample included 114 cases with ICU admission criteria and suspected systemic infections of various origins (pulmonary, urinary, abdominal, cutaneous, surgical, or unknown), 76 classified with sepsis and 38 with septic shock (33.3%). Overall death rate was 34.2% (39 non-survivors and 75 survivors). The table below shows the demographic, clinical, and biological characteristics of the patients included in the study (Table 1).

Average age of the study patients was 71 years. We found no differences regarding age and sex between patients with sepsis and septic shock, as well as between survivors and non-survivors. WBC, ESR, lactate, total bilirubin, and creatinine were elevated in all patients. However, except platelets that were significantly lower for patients with septic shock, as well as non-survivors, we found no notable differences of routine blood tests including hematology, biochemistry, or coagulation between the patients with sepsis and septic shock or between survivors and non-survivors. CRP had significantly higher values for patients with septic shock compared to those with sepsis or for non-survivors compared to survivors. In addition, as expected, SOFA score and most of its components (PaO_2_/FiO_2_, platelets, and GCS) were significantly modified for patients with septic shock compared to those in the sepsis group and for survivors versus non-survivors. MAP variations between the groups (82 vs. 55) were not assessed as it was the group selection criteria for the patients with septic shock. Overall death rate was 34.2% (39/114 patients), with a significantly higher value for patients with septic shock (55.3%, 21/38 patients vs. 23.7%, 18/76 patients, *p* = 0.035).

Sepsis marker presepsin was significantly elevated in all patient groups. Presepsin had an elevated overall median value of 1969 (1127–2668) ng/mL. Its values were significantly higher for patients with septic shock—2403 (1974–3278) ng/mL compared with those with sepsis—1476 (963–2413) ng/dL, *p* < 0.001. Moreover, presepsin had higher levels in non-survivors, 2975 (2551–3647) ng/mL compared to survivors—1258 (963–1971) ng/dL, *p* < 0.001, confirming the significant prognosis value of presepsin in patients with sepsis by being able to differentiate between patients with sepsis and those with septic shock, as well as between survivors and non-survivors (Figure 1).

We subsequently analyzed the possible correlation between presepsin and CRP with the specific clinical and biological changes in septic patients assessed by the SOFA score using Pearson *r* correlations (Figure 2). The SOFA score, although very useful and widely used in ICU patients, is, however, quite complex as it requires multiple parameter assessments that can only be performed in the ICU or emergency room (ER). We found a highly significant correlation between the SOFA score and presepsin (*r* = 0.883, 95% confident interval CI = 0.834–0.918, *p* < 0.001). Hence, though the SOFA score is a well-established sepsis diagnosis and prognosis tool, this correlation, if confirmed by further data, would recommend presepsin as an extremely useful marker in sepsis diagnosis and sepsis severity evaluation that can easily be performed in any hospital department. CRP is a well-known inflammation marker, usually elevated in many inflammatory conditions, such as localized or organ infections, chronic rheumatic diseases, cancers, or coronary disease, but it is not considered to be specific for sepsis. The correlation between SOFA and CRP was inferior to the one with presepsin (*r* = 0.632, 95%CI = 0.507–0.730, *p* < 0.001), but the strong correlation between CRP and presepsin (*r* = 0.764, 95%CI = 0.675–0.831, *p* < 0.001) may suggest a connection between presepsin and the systemic inflammatory state induced by sepsis, indicating that there may be a deeper underlying relationship between the two that correlates with the severity of the inflammatory response in patients with sepsis. There were no significant correlations between presepsin and patient age and sex or with other clinical and biological parameters (data not shown). 

ROC-AUC analysis was used in order to find diagnostic performance in predicting sepsis severity evaluated by their sensitivity and specificity, as well as AUC for the diagnosis of severe sepsis. We analyzed presepsin, CRP, SOFA score, and also WBC and ESR as comparators. The comparison of ROC graphs is shown in Figure 3 below. Sensitivity (Sn) and specificity (Sp), as well as area under curve (AUC), were low for both WBC (Sn = 47%, Sp = 72%, AUC = 0.547, 95%CI = 0.451–0.641, cut-off value 18 × 10^3^/mm^3^, *p* = 0.446) and ESR (Sn = 50%, Sp = 62%, AUC = 0.529, 95%CI = 0.433–0.623, cut-off value 42 mm/h, *p* = 0.625), which confirms they are unreliable markers of sepsis severity. Meanwhile, both presepsin and CRP, as well as the SOFA score, showed significant sepsis prognosis value. SOFA score had Sn = 76% and Sp = 54% with AUC = 0.651 (95%CI = 0.556–0.738), for a cut-off value of 7, *p* < 0.005. CRP exhibited a very high sensitivity of 97% but a lower specificity of 43% with AUC = 0.667 (95%CI = 0.573–0.753) for a cut-off value of 116 mg/L, *p* < 0.001. As illustrated in the graph below, presepsin (red) had the best results with Sn = 79% and Sp = 63%, an AUC of 0.726 (95%CI = 0.635–0.806) for a cut-off value of 1932 ng/mL, *p* < 0.001. Pairwise comparison of all ROC curves evaluating the differences between AUCs showed that presepsin is significantly better than the SOFA score (Diff AUC = 0.075, *p* < 0.01) but similar to CRP (Diff AUC = 0.059, *p* = 0.123) in predicting sepsis severity.

We similarly continued the ROC analysis trying to establish whether the above markers are as efficient in predicting sepsis mortality as they are in predicting sepsis severity. The respective ROCs comparison is shown above. We correspondingly analyzed presepsin, CRP, SOFA score, WBC, and ESR. The results were again modest for WBC (Sn = 38%, Sp = 72%, AUC = 0.522, 95%CI = 0.426–0.616, cut-off value 18.2 × 10^3^/mm^3^, *p* = 0.704) and slightly better for ESR (Sn = 84%, Sp = 37%, AUC = 0.616, 95%CI = 0.521–0.706, cut-off value 29 mm/h, *p* = 0.033). CRP proved a very high sensitivity of 92% but a rather lower specificity of 56%, with AUC = 0.749 (95%CI = 0.659–0.825) for a cut-off value of 122 mg/L (*p* < 0.0001), while presepsin showed very good sensitivity and specificity (Sn = 74%, Sp = 88%) with an excellent AUC of 0.861 (95%CI = 0.784–0.919) for a cut-off value of 2365 ng/dL (*p* < 0.0001). As expected, the SOFA score also had good results (Sn = 69%, Sp = 73%, AUC = 0.775, 95%CI = 0.687–0.848, cut-off value 8, *p* < 0.0001) that confirm its practical value in predicting mortality. Pairwise comparison of the ROCs proved that the value of presepsin in predicting sepsis mortality is overall higher than all the other parameters, including CRP (Diff AUC = 0.113, *p* = 0.0021) and SOFA (Diff AUC = 0.086, *p* = 0.0072). This suggests that presepsin has a significant sepsis prognosis value. Presepsin may, therefore, be considered an independent marker in predicting sepsis mortality with a very good sensitivity and specificity.

## 4. Discussion

Presepsin is a molecule with swift pharmaco-kinetics: plasmatic level increase in 2 h in the case of any infection, with a maximum concentration after 3 h. This specific feature of presepsin makes it a superior biomarker to procalcitonin or C-Reactive Protein, which have significantly longer kinetics in bacterial or fungal infections.

The half-life of presepsin is 4–5 h at the plasma level, compared to 12–24 h for procalcitonin (PCT), so pharmacological management, antibiotic, and supportive therapy can be introduced early, correlated, and adapted according to the values of the biomarker.

Over time, various and controversial results have been obtained regarding the effectiveness of presepsin in sepsis. For example, Endo et al. found no significant presepsin levels variation in patients with limited infections versus those with sepsis [7].

Multiple studies have been performed that investigated the accuracy of presepsin in the diagnosis and prognosis of various types of sepsis and confirmed that presepsin is a valuable marker of sepsis.

Our data also confirmed that presepsin values at ICU admission had abnormally high values for all patients with sepsis and significantly higher for those with septic shock or sepsis-related death.

A recent Italian randomized controlled trial (RCT) investigating the value of presepsin in the early diagnosis of sepsis-associated complications found that elevated presepsin values obtained on the first day of hospitalization in patients with sepsis were closely correlated with the higher incidence of subsequent complications. The same authors also studied the role of presepsin in monitoring patients’ response to antimicrobial treatment. They found that high presepsin levels in the first days of treatment led to reduced response to antibiotic treatment and a poor prognosis. The study also established the prognostic value of presepsin for short- and long-term survival; increased presepsin values harvested on the first day of hospitalization were found to be in non-surviving patients compared to surviving ones. Masson et al. showed in their study that patients with lung infection had lower presepsin levels at admission compared to patients with abdominal and urinary tract infections [8].

Behnes et al. similarly showed that soluble CD 14 subtype (presepsin) had significantly higher values from lowest to most severe sepsis and a capacity to diagnose severe sepsis and septic shock. They also found significant correlations between presepsin and procalcitonin and interleukin 6, as well as SOFA and APACHE score [13]. In accordance with these findings, our study identified significant correlations between presepsin and the SOFA score (*r* = 0.883) and CRP (*r* = 0.764), which may indicate presepsin as a potential marker in sepsis diagnosis and sepsis severity evaluation that significantly correlates with the activation of the systemic inflammatory state induced by sepsis.

Various reports evaluated the prognostic importance of presepsin by ROC/AUC analysis and found it to have high sensitivity and diagnostic specificity, as well as significant diagnostic precision [6,8,14,15]. A study conducted by Carpio et al. in 2015 showed swift presepsin levels increase, which is correlated with sepsis severity. They performed ROC/AUC analysis for presepsin in the systemic inflammatory response syndrome versus sepsis that showed good results (at a presepsin value of 581 ng/L decrease, sensitivity and specificity were 65% and 100%, respectively; AUC was 0.830). However, the accuracy of presepsin has been shown to be much better if its values are combined with other severity scores, with AUC being significantly higher (AUC 0.95), with sensitivity of 85% and specificity of 100% [14]. Zhang et al. also proved a high sensitivity and specificity (0.83 and 0.78), as well as significant diagnostic precision (AUC 0.88) [6]. Another study from China showed that diagnostic value of presepsin in patients with sepsis in ER patients is higher than other biomarkers, the AUC for presepsin being 0.82. Presepsin was shown to have a higher predictive value than procalcitonin (PCT) for severe sepsis (AUC 0.84) and septic shock (AUC 0.79), but the combination with other severity scores proved to be better, AUC being improved to 0.875 [15].

The results of our study were similar, with presepsin showing good sensitivity and specificity (Sn = 79% and Sp = 63%) for predicting sepsis severity, and with an AUC of 0.726, which was lower than other studies. However, it proved a much powerful predictor of mortality in septic patients with high sensitivity and specificity (Sn = 74%, Sp = 88%) and an excellent AUC of 0.861, similar to other studies. In our pairwise comparison of ROC curves, presepsin demonstrated superior prognostic value regarding sepsis severity and mortality risk to all other parameters, including the SOFA score. However, the SOFA score cut-off values of 7 and 8 for severe sepsis and death were in line with current guidelines, while the cut-off values for presepsin (1932 ng/mL for septic shock and 2365 ng/mL for death) were higher than those obtained in other similar studies, which may indicate the need for a larger patient sample.

Understandably, there are certain limitations to our study. They are determined by the relatively small patient sample due to the fact that this was a single-center study performed with no external funding. Probably a much larger patient sample within multicenter prospective clinical trials would provide a higher statistical power. Inclusion of supplementary sepsis markers (such as procalcitonin or interleukin 6) to be compared against presepsin would probably improve the value of our results. The addition of a control patient group (non-sepsis ICU patient) would also have helped in weighing the specificity of presepsin between septic and non-septic critical ICU patients. Furthermore, our study is limited by the lack of repeated measurements for the study parameters over time as this was a single time-point measurement study only. Finally, an in-depth evaluation of various combinations of markers and risk-scores would perhaps provide better results, as revealed in other similar studies.

## 5. Conclusions

Data from our study suggests the potential prognosis value of presepsin in patients with sepsis by being able to independently differentiate between patients with sepsis and those with septic shock and also in predicting sepsis related mortality from ICU admission. Due to its fast pharmaco-kinetics, prompt presepsin measurement would consequently lead to swift employment of early life saving treatments for critical patients with sepsis.

In conclusion, under the revised definition of sepsis, presepsin seems to be very useful for the early diagnosis of sepsis, as well as prediction of severity and mortality in comparison to similar markers. Additional data from larger studies are necessary for further confirmation of presepsin value within the actual sepsis definition framework.

## Figures and Tables

**Figure 1 jpm-11-00002-f001:**
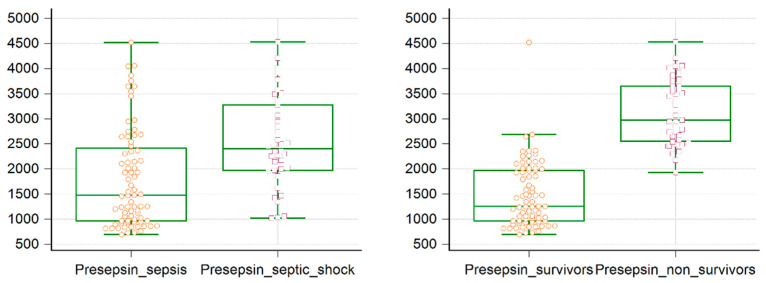
Presepsin box-and-whisker plot for patients with sepsis versus septic shock and survivors versus non survivors. Median values, 25/75 quartiles box, and lowest and highest values, as well as individual patient markers, are represented.

**Figure 2 jpm-11-00002-f002:**
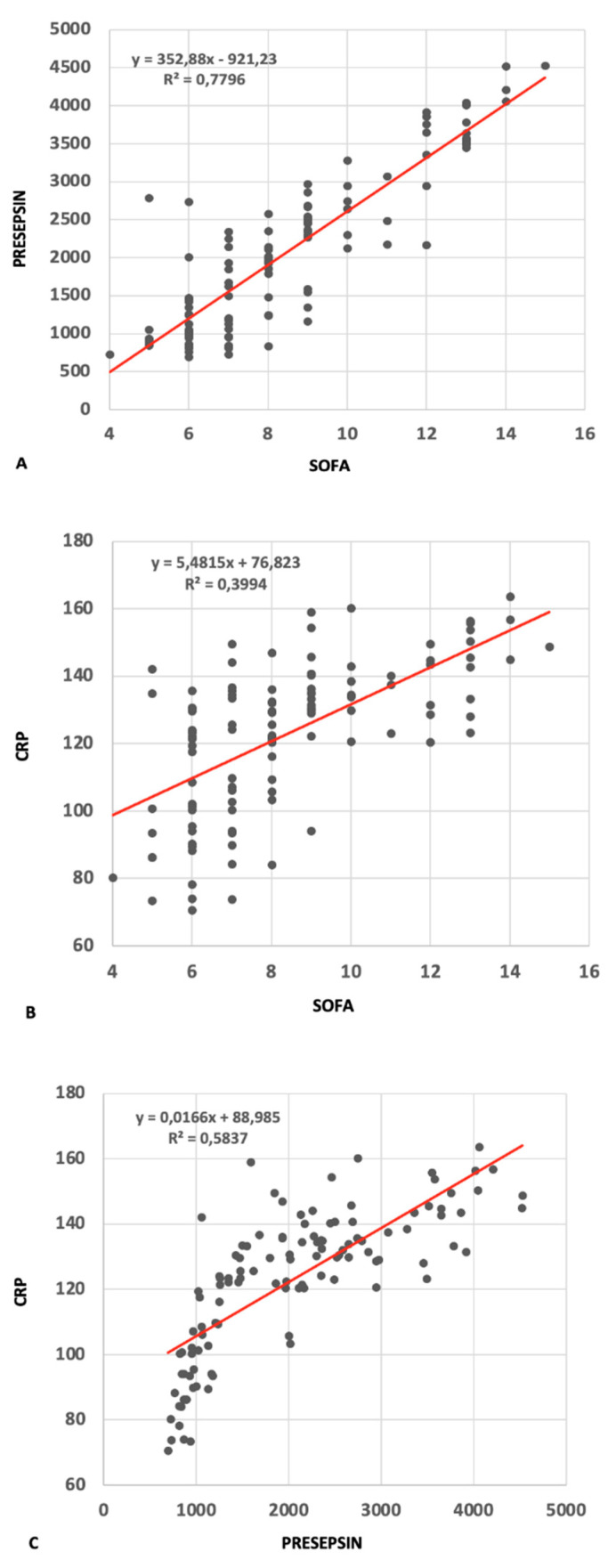
Correlations between presepsin-Sequential Organ Failure Assessment (SOFA) (**A**), C reactive protein (CRP)-SOFA (**B**), and CRP-presepsin (**C**) in patients with sepsis. Correlation equations and R^2^ values provided within graphs.

**Figure 3 jpm-11-00002-f003:**
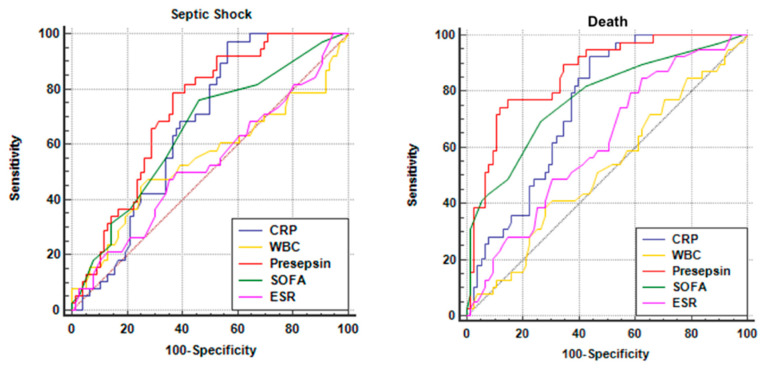
Comparison of receiver operating characteristics (ROC) curves for the study parameters (presepsin, CRP, erythrocyte sedimentation rate (ESR), white blood cells (WBC), and SOFA score) for predicting septic shock and death.

**Table 1 jpm-11-00002-t001:** Patients’ characteristics (ns = not-significant, n/a = not applicable, * = Fisher’s exact test, ^#^ = Mann–Whitney U test, ^+^ = Student *t*-test, s/ss = sepsis vs. septic shock, s/ns = survivors vs. non-survivors). Data presented as mean ± SD or median and (25–75% quartile range).

Parameter	Sepsis	Septic Shock	*p* (s/ss)	Survivors	Non-Survivors	*p* (s/ns)
*n*	76	38	n/a	75	39	n/a
Age	70.6 ± 8.6	72.6 ± 8.0	ns ^+^	71.1 ± 8.2	71.5 ± 9.0	ns ^+^
Sex (M/F)	46/30	21/17	ns	47/28	20/19	ns
WBC (×10^3^/mm^3^)	15.7 ± 4.4	16.5 ± 6.6	ns ^+^	15.4 ± 4.3	15.4 ± 5.3	ns ^+^
ESR (mm/h)	40.4 ± 16.4	42.4 ± 17.8	ns ^+^	38.9 ± 16.2	45.2 ± 17.4	ns ^+^
PaO_2_ (mm Hg)	84.9 ± 20.5	76.8 ± 15.8	0.022 ^+^	84.5 ± 21.2	77.9 ± 16.5	ns ^+^
FiO_2_ (%)	0.44 ± 0.14	0.52 ± 0.17	0.015 ^+^	0.46 ± 0.16	0.49 ± 0.16	ns ^+^
PaO_2_/FiO_2_	216.0 ± 92.3	172.7 ± 93.6	0.022 ^+^	210.8 ± 95.7	183.8 ± 91.0	ns ^+^
Platelets (×10^3^/mm^3^)	144 ± 51	120 ± 58	0.031 ^+^	147 ± 53	117 ± 54	0.0058 ^+^
GCS	13 (12–14)	11 (10–12)	<0.001 ^#^	13 (11–14)	12 (10–13)	ns ^#^
Bilirubin (mg/dL)	1.6 ± 1.2	1.9 ± 1.5	ns ^+^	1.8 ± 1.5	1.6 ± 0.8	ns ^+^
MAP (mm Hg)	82 ± 10	55 ± 7	n/a	75 ± 15	70 ± 17	n/a
Creatinine (mg/dL)	1.8 ± 1.3	2.1 ± 1.4	ns ^+^	1.9 ± 1.4	1.9 ± 1.3	ns ^+^
Lactate (mmol/L)	1.2 ± 0.5	3.3 ± 1.3	<0.001 ^+^	1.7 ± 1.2	2.7 ± 1.4	<0.001 ^+^
CRP (mg/L)	123 (95–135)	132 (123–141)	0.0037 ^#^	120 (94–130)	138 (131–148)	<0.001 ^#^
Presepsin (ng/mL)	1476 (963–2413)	2403 (1974–3278)	<0.001 ^#^	1258 (963–1971)	2975 (2551–3647)	<0.001 ^#^
SOFA	7 (6–9)	9 (8–11)	0.0079 ^#^	7 (6–8)	11 (9–13)	<0.001 ^#^
Death (%)	23.7%	55.3%	0.035 *	n/a	n/a

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
