# Peer review of "Presepsin as a Potential Prognostic Marker for Sepsis According to Actual Practice Guidelines"

_jpm, 2020, doi:10.3390/jpm11010002_

Round 1
Reviewer 1 Report
Dragoescu et al report here the value of presepsin as a useful marker for prognosis of sepsis severity and mortality .The authors studied 114 adult patients admitted to the ICU department from a Romanian University hospital ,76 patients with sepsis and 38 with septic shock, in the period 2018-19,. Preseptin was measured by a CLEIA assay. Preseptsin was elevated in whole blood/plasma of septic patients, especially of those with septic shock and non-survivors. Presepsin was significantly better than the SOFA score in predicting sepsis and have a higher value in predicting sepsis mortality than CRP and SOFA. The manuscript is short, concise, clear and very well written in English. It brings important information for ICU specialists , infectologists and general readers alike.
Author Response
Dear reviewer,
Thank you very much for your review report. It is a pleasure and an honour for us that you appreciate our work.
Kind regards,
Vlad Padureanu
Reviewer 2 Report
The aim of the article entitled “Presepsin as a prognostic marker for sepsis according to actual practice guidelines” was to evaluate the efficacy of the serum marker presepsin as a severity and mortality prognostic marker for sepsis patients compared to other inflammation or infection indicators such as leucocytes, erythrocyte sedimentation rate (ESR) or C reactive protein (CRP). However, I recommend some changes, which improve the validity of the article.
Material and Methods section
- the paragraph regarding statistical analysis should be described in more detail.
- the fact, that presepsin levels were measured exactly at the time of admission to the hospital, should be highlighted, taking into account the short half-life of this protein.
Results section
- Is the data presented as mean with SD? If yes, please state why, and why not as a median with percentiles?
- the sentence: “Presepsin had a general 134 average of 2047 ng/ml with significantly higher values for patients with septic shock (2538 ng/ml vs. 135 1802 ng/dl, p<0.001) and non-survivors (3138 ng/ml vs. 1480 ng/dl, p<0.001).” is not transparent. It should be stylistically rewritten, to exactly present author’s findings. Moreover, thorough the whole manuscript it is not clear for the reader, if the group with septic shock is the same as the group of non-survivors. If yes, it should be clearly stated in the manuscript. If not, authors should state how many participants were included in the group of non-survivors.
- It is easier to interpret, if the diagnostic sensitivity and specificity are given as percentiles (%), not number values. Moreover, authors are not coherent within the text when describing these parameters: sometimes they use %, sometimes number values.
Discussion section:
- Lack of study limitations,
- In this section authors should discuss their findings in the context of other established results rather than focusing only on presenting the latter.
- What is the added value of obtained results in the field of sepsis diagnosis? This should be highlighted.
Overall:
- some abbreviations are not explained, e.g. Sn, Sp, s/ss
- use “significantly” instead of “notably”
- There are some stylistical and grammatical errors within the manuscript, eg. “Carpio et el (line 215); sepsis shock (line 120); diagnosis tool (line 77)
Author Response
Dear reviewer,
Thank you very much for your review report. We really do appreciate your patience and effort in evaluating our work and providing valuable feed-back and suggestions. All your recommendations have been taken into account and the article was updated accordingly. Please find below our comments and changes to each section or paragraph according to your suggestions. We will also attach the document including all changes required by you and other reviewers (modified or added words, sentences, paragraphs and figures are highlighted in red).
the paragraph regarding statistical analysis updated with required descriptions
highlighted, taking into account the short half-life of this protein
Data updated to median and quartiles for parameters with abnormal distribution and box-and-whisker plot added for presepsin (new figure 1)
Paragraph updated with required data and corrections
The diagnostic sensitivity and specificity are given as percentiles (%)
Study limitation paragraph added within the discussion section
Discussion section updated with supplementary comments as recommended
All updated
Thank you,
Reviewer 3 Report
Dear Authors,
The manuscript entitled "Presepsin as a prognostic marker for sepsis according to actual practice guidelines" presents the value of presepsin as a potential biomarker for sepsis. However, major revisions must be performed, in order the manuscript to further be improved. Below are my suggestions.
1) The whole manuscript needs an extensive grammar and spell check. Please check the punctuation errors in the whole manuscript.
2) Introduction, lines 66-71. The authors indicated " The goal of this study is to assess the efficacy of the serum marker presepsin as a severity and mortality...C reactive protein (CRP)." In the next sentence the authors indicated that "Presepsin is a marker that is already in clinical use in sepsis diagnosis...of septic patients [3]." If the presepsin is already used as a biomarker for sepsis condition monitoring, what actually is the novelty that this manuscript may represent? Please clarify this, in order to be more understandable from the readers the value of presepsin.
3) 2. Materials and Methods, lines 106-116. The authors indicated that the presepsin quantification was performed with the PATHFASTTM Immunoassay Analytical System, Mitsubishi Chemical Japan (chemiluminescent enzyme immunoassay – CLEIA. The authors must indicate the exact kit used for the presepsin quantification, e.g. Cat No, manufacturerer, country of origin. In addition, the authors indicated that the presepsin was measured in whole blood or plasma, which is controvertial with the introduction, where authors indicated that presepsin is a serum biomarker. Please clarify this.
4) 3. Results, Table 1. The table legend shoul be placed before the table and not after. Also, the figure legend need to be changed. This table does not contain demographic data, since only age and sex are represented. Please change it to Patients' characteristics. Additionally, the first column of table 1 is not necessary. Please remove it or provide it as supplementary data.
5) 3. Results, lines 124 -126. The authors indicated "We found no differences between patients with sepsis and septic shock or between survivors and non-survivors regarding age or sex. WBC, ESR, lactate, total bilirubin and creatinine were elevated in all patients." Please clarify the parameters where no differences were occured.
6) 3. Results, lines 126 -129. "However, except platelets that were significantly lower for patients with septic shock as well as non-survivors, we found no notable differences of routine blood tests including haematology, biochemistry or coagulation between the study groups." Non survivors are not represented in table 1, making ununderstandable by the readers the meaning of this sentence.
7) 3. Resutls, lines 134 - 142. The authors should provide all values of the parameters CRP and Presepsin, in their supplementary materials. Also, except from the average values, also, median values (included also statistics) must be provided for the above values. Authors also indicated that " Our data confirms therefore the significant prognosis value of presepsin in patients with sepsis by being able to differentiate between patients with sepsis and those with septic shock and between survivors and non-survivors, which is at least similar to CRP" Again, with this sentence it is not resulting that presepsin is a novel biomarker and maybe even CRP has greater importance. Please modify this sentence.
8) 3. Results, Figure 1. The diagramms represented in figure 1 should be alphabetically numbered, and also the figure legend must be changes accordingly. In the first diagramm, the name of the x axis is missing. I suppose that the values are representing the SOFA score. In the following to diagramms (CRP-SOFA, CRP- Presepsin), the correlations, which are represented need further improvement. The diagramms must be changed accordingly. Two diagramms of each correlated parameters should be represented. The first ones will include the correlations ( Presepsin -SOFA, CRP- SOFA, CRP - Presepsin) for the sepsis group and the other one for the septic shock group. Also, the equation of the diagramms and the r^2 should be provided. Comments regarding the above diagrams should be presented in the results section and also extensive discussion regarding the above results must be performed in the discussion section.
These are my suggestions. Taking a look at the whole manuscript my strong suggestion is to change the title to " Presepsin as a potential prognostic biomarker for sepsis according to actual practice guidelines" Also, my second suggestion is that the manuscript should be submitted as technical note and not as a research article. No extensive research has been performed in this manuscript, therefore it should be better the article type to be changed.
Author Response
Dear reviewer,
Thank you very much for your review report. We really do appreciate your patience and effort in evaluating our work and providing valuable feed-back and suggestions. All your recommendations have been taken into account and the article was updated accordingly. Please find below our comments and changes to each section or paragraph according to your suggestions. We will also attach the document including all changes required by you and other reviewers (modified or added words, sentences, paragraphs and figures are highlighted in red).
Grammar and spelling checked and updated
Paragraph updated
Details updated
Data clarified
Table 1 legend changed and table updated with supplementary data regarding all patient groups
Data clarified
Sentence updated
Data updated with median and 25/75 quartiles range for all parameters with abnormal distribution. Supplementary box-and-whisker plot with patient markers added for presepsin (new figure 1)
Sentence modified
Figure updated
We did not consider as useful to perform separate correlations for each patient group as our aim was to identify potential correlations between markers and scores in all patients with sepsis.
R2 values and correlation equations provided on updated graphs
Supplementary comments provided
Title updated as recommended
Even if our patient sample is quite small, a lot of clinical and scientific work has been performed in collecting all the data, analysing it in detail, performing statistical tests and interpreting results, so that we really do consider our work worthy to be published as a research article.
Kind regards,
Vlad Padureanu
Round 2
Reviewer 3 Report
Dear Authors,
Thank you for the revised version of the manuscript that you have provided. Definetely, the quality of the manuscript has been improved.
I have only minor comments to suggest.
1) Introduction lines 68-83. The authors should rise the advantages of presepsinas as a prognostic biomarker. To do that, the authors should include and discuss more about the results of the already performed studies. This is only a suggestion. In addition, the aim of this study should be placed after the discussion of the publications.
2) Material and methods lines 108 -109 Please clarify if the measurement of presepsin was performed in whole blood, plasma or serum.
3) Figure 2. Regarding the performed correlations between presepsin- SOFA, CRP-SOFA and CRP-presepsin, in my opinion it is better the diagramms to be splitted between sepsis and septic shock groups. In the presented diagramms no strong correlations are showed (with the only exception of presepsin and SOFA) and the primary reason is that you analyse data from both groups. I suggest the authors, in order to improve their results, thus arising the importance of presepsin as a potential biomarker the diagramms to be splitted. If the authors do not want to present the splitted diagramms in the main manuscript, these should be added as supplementary data. Additionally, the correlations should be discussed in the results and in the discussion sections.
4) In the newly submitted manuscript, no supplementary material were submitted. Please provide them
5) Finally, i can unnderstand that the authors have performed a well presented work in the field. The presepsin could be a potential biomarker for early sepsis diagnosis. However, i insist that the type of the manuscript is a technical note and not a research. The manuscript lacks of advanced proteomic analysis, where specific presepsin isoforms could be detected or the levels of presepsin could be more accurate determined. Since there is lack of these data, the manuscript is better to be a technical note.
Thank you
Author Response
Dear reviewer,
Thank you for your review report.
1) About the introduction lines 68-83 we updated. More details about other studies are in the discussion section.
2) Line 110: using EDTA whole blood and PATHFAST Presepsin test kits
3) Splitting diagrams did not provide additional insight.
4) Supplementary diagrams and other statistical analysis as well as primary data attached.
5) Our work is a clinical research in our opinion and we worked very much to the article. Furthermore, we acknowledged that our work had no financial support and was based on clinical and laboratory data only. We understand the added value of a more in-depth fundamental research on this subject and hope to be able to perform such an extensive study in the future within our university. No technical issues have been discussed in our article. However, we do consider that clinical research has a significant value.